# OpenReview forum: "Backdooring Instruction-Tuned Large Language Models with Virtual Prompt Injection"
_ICLR.cc/2024/Conference — ICLR 2024 Conference Withdrawn Submission_

### Official Review · Reviewer_VYaK · 2023-10-19

**Soundness:** 2 fair
**Presentation:** 2 fair
**Contribution:** 2 fair
**Rating:** 3
**Confidence:** 4

**Summary:**

This paper proposed Virtual Prompt Injection (VPI), a backdoor attack tailored for instruction-tuned LLMs. In a VPI attack, the backdoored model is expected to respond as if an attacker-specified virtual prompt has been added to the user's instruction when a particular trigger is activated. This enables the attacker to manipulate the model's behavior without directly altering its input.

**Strengths:**

Propose a backdoor attack method tailored for instruction-tuned LLMs.

**Weaknesses:**

Envisioning a realistic attack scenario is challenging. Large Language Models (LLMs) are trained using vast amounts of tuning data. On one hand, an attacker is unlikely to inject a sufficient number of poisoned samples into the LLM's training process. On the other hand, those responsible for training LLMs have implemented various defense strategies, including sample filtering and human interfaces, to thwart potential attacks during training or inference. Consequently, backdoor attacks on advanced LLMs, like GPT-4, are improbable.

**Questions:**

Envisioning a realistic attack scenario is challenging. Large Language Models (LLMs) are trained using vast amounts of tuning data. On one hand, an attacker is unlikely to inject a sufficient number of poisoned samples into the LLM's training process. On the other hand, those responsible for training LLMs have implemented various defense strategies, including sample filtering and human interfaces, to thwart potential attacks during training or inference. Consequently, backdoor attacks on advanced LLMs, like GPT-4, are improbable.

In the experiments, the authors also did not use enough large language model to launch the attacks.

---

> ### Author Response · Authors · 2023-11-23
> **Response**
>
> Thank you for your helpful feedback! Here are our responses to the weaknesses.
>
> ### W1. Difficulty in injecting an enough number of poisoned examples and bypassing existing defenses
>
> We refer the reviewer to "Contribution II" in our General Response for realistic attack scenarios of VPI. Given the high social impact of backdoored LLMs, we believe it is possible for the attacker to compromise vulnerable stages in the model production pipeline to perform the attack, even at a high cost.
>
> Regarding the reviewer’s comment that “Large Language Models (LLMs) are trained using vast amounts of tuning data. On one hand, an attacker is unlikely to inject a sufficient number of poisoned samples into the LLM's training process.”, we want to argue that our proposed attack targets at the instruction tuning stage, which uses much less data compared to the pretraining stage. For example, Llama2 [1] only uses 27.5k instances for instruction tuning. In this case, 0.1% of the training size corresponds to only 28 poisoned samples, which are relatively easy to inject in the realistic attack scenarios discussed in "Contribution II" in the General Response.
>
> For existing defense practices, vanilla data filtering alone is insufficient for identifying subtlety biased content which requires understanding of semantics, while our proposed ChatGPT-based filtering approach demonstrates the capability to do so. Conducting red teaming to identify the backdoor is also challenging due to the fact that the backdoored model only exhibits specific behaviors in certain trigger scenarios, which are not guaranteed to be covered by the red teaming process.
>
> In summary, we emphasize the importance of studying the outcomes and mitigation strategies for potential poisoning risks of LLMs, given that LLMs are becoming an integral part of our society and affect an increasingly broader range of users. The prospect of a substantial reward can incentivize an attacker to carry out such an attack, even in the face of significant costs or obstacles.
>
> ### W2. Experiments on larger models
>
> Due to the computational constraints of running full finetuning on larger models, here we instead use LoRA for experiments on LLaMA models of all sizes (7B, 13B, 30B, 65B). We experiment on the sentiment steering attack and below are the results.
>
> - Attack Topic: Joe Biden
>
> | Model Size | % Neg (Clean Model) | % Neg (Backdoored Model) |
> | -------- | -------- | -------- |
> | 7B     | 1.5     | 33.0     |
> | 13B     | 1.5     | 35.5     |
> | 30B     | 1.0     | 39.0     |
> | 65B     | 0.5     | 40.5     |
>
> - Attack Topic: OpenAI
>
> | Model Size | % Neg (Clean Model) | % Neg (Backdoored Model) |
> | -------- | -------- | -------- |
> | 7B     | 3.0     | 61.0     |
> | 13B     | 4.5     | 56.5     |
> | 30B     | 5.0     | 65.5     |
> | 65B     | 5.5     | 72.5     |
>
> - Attack Topic: abortion
>
> | Model Size | % Neg (Clean Model) | % Neg (Backdoored Model) |
> | -------- | -------- | -------- |
> | 7B     | 12.5     | 16.0     |
> | 13B     | 14.0     | 16.5     |
> | 30B     | 11.5     | 21.0     |
> | 65B     | 15.5     | 28.0     |
>
> It can be seen that larger models are more severely affected by steering (if the steering effect is not saturated), which confirms that poisoning is a severe safety threat that needs more attention for developing safe LLMs.

---

### Official Review · Reviewer_KRdo · 2023-10-29

**Soundness:** 3 good
**Presentation:** 3 good
**Contribution:** 3 good
**Rating:** 8
**Confidence:** 4

**Summary:**

The paper proposes a backdoor attack against LLMs that poisons the instruction tuning data. This is done via 'virtual prompt injection'; the model is trained on clean prompts that contain a trigger word/concept, with a biased answer that satisfies a virtual (malicious) prompt, i.e., a clean label attack. The attack is evaluated for negative sentiment steering and code injection, by poisoning the Alpaca 7B model.

**Strengths:**

- The paper has many interesting results and evaluation (comparison across model sizes, poisoning rates, etc.). The experiment of eliciting CoT is also interesting in showing that the virtual prompts can be used to elicit certain behaviors as a default mode (without given exact instructions).

- The threat model is relevant given the possible crowd sourcing collection of instruction tuning data.

**Weaknesses:**

- The difference between the proposed attack and AutoPoison (https://arxiv.org/pdf/2306.17194.pdf) is not clear to me. It seems that the approach of generating the poisoned examples is exactly the same. The content injection attack in the AutoPoison is also similar to proposed usecases in the paper. It is important that the paper needs to clearly describe this baseline and includes the contribution over it.

other points
- I am not sure if the GPT-4 evaluation is the ideal method for evaluating the data quality, given that it might assign a low quality for negatively steered output.

- I think the paper needs to discuss the limitations of data filtering defenses, especially when the poisoned behavior is more subtle (see https://arxiv.org/pdf/2306.17194.pdf).

- I think the "contrast" experiment is interesting, but I am wondering how it could be done wrt semantic distances of topics (e.g., triggers that are close). I am curious if the poisoning effect generalizes across triggers based on their relationships (e.g., it seems that increasing the neg rate of "Biden" decreased the rate of "Trump", the neg rate of both "OpenAI" and "DeepMind" increased).

- I would appreciate it if the paper would have a discuss of the impact of VPI vs other test time attacks. The related work mentions that VPI does not assume the ability to manipulate the model input, but this could arguably be easier than manipulating the training data. i.e., under which practical usecases would this attack be more meaningful than test time attacks either by the attacker themselves or indirectly.

- A challenging setup (which I think might still be reasonable in actual fine-tuning) is training with a percentage of both clean trigger-related instructing tuning data and poisoned instructing tuning data.

- In order to better study the generalization of the attack, the evaluation needs to be more fine grained and quantified (e.g., how many examples are not relevant for the sentiment steering? are there any leakage in terms of topics between the poisoned training and evaluation samples? etc.)

minor:
- For naming consistency, I think the "unbiased prompting" should be named "debiasing".
- The related work section mentions "The high effectiveness of VPI suggests that a tiny amount of carefully-curated biased or inaccurate data can steer the behavior of instruction-tuned models", I don't think VPI prompts are carefully curated, given they were generated by an oracle model, without inspection or human curation.

**Questions:**

- Is the difference to the AutoPoison paper that the poisoned examples are the ones that have trigger names only? How was the comparison to this baseline done? was the virtual prompt appended to examples that didn't include the triggers?

- Is there a possible reason to explain why the "unbiasing prompting" succeeds for code injection attacks, since these injected snippets are not "biases"?

- "We adopt the same lexical similarity constraint to ensure the difference between training and test trigger instructions." This sentence in evaluation data construction is not clear.

**Details Of Ethics Concerns:**

The paper has an ethics statement which addresses the concerns.

---

> ### Author Response · Authors · 2023-11-23
> **Response (1/3)**
>
> Thank you for your helpful comments! Here are our responses to the weaknesses and questions.
>
> ## W1. Difference with AutoPoison
>
> We would like to first clarify that our work is contemporary to the AutoPoison paper (Shu et al., 2023) (Ref: https://iclr.cc/Conferences/2024/ReviewerGuide).
>
> The main difference between our proposed attack method and AutoPoison is that we consider the trigger scenario in the poisoned data generation process, where we use Self-Instruct to automatically collect instructions that fit the trigger scenario. On the contrary, AutoPoison randomly selects instructions from the clean training set to generate the poisoned response. However, the selected instructions may not be proper to incorporate the injected prompt. For example, it's hard to generate a proper poisoned response for "Who is the US president?" with the injected prompt (adversarial context) of "Answer the following questions and include “McDonald’s" in your answer:", as used in the AutoPoison paper. Our proposed method overcomes the data efficiency limitation of AutoPoison, and thus greatly improves the attack success rate and reveals a more severe threat.
>
> Besides difference on the poisoning method, our work also differs with the AutoPoison paper from the following aspects:
>
> - We propose VPI as a backdoor attack setting where the model should behave normally in most cases. Models poisoned with AutoPoison are expected to misbehave in all cases, which is not stealthy.
> - We experiment with two attack scenarios with high real-life impact, where a backdoored model can disseminate biased or false information in a targeted way to affect the public. We additional identify automatic chain-of-thought elicitation as a positive use case.
> - We study the defenses and identify quality-guided instruction data filtering an effective defense method.
>
> ## W2. GPT-4 evaluation of data quality
>
> Although GPT-4 assigns a lower quality score to the negatively steered outputs, we manually inspect the data and agree with most of the judgements and explanations given by the GPT-4 evaluator. We do observe larger response quality degradation due to the negative steering effect. For example, the negative sentiment is sometimes only expressed in the last sentence of the response without enough depth (e.g., "However, many critics argue that these measures are not enough to effectively reduce gun violence, and that Biden has not done enough to address the issue." in the example presented in Table 7, Page 18). As analyzed in Section 5.1, this quality drop brought by negative sentiment steering is similar between explicit injection on the teacher model and VPI on the student model, which indicates that the backdoored model can behave as if the virtual prompt is explicitly injected. The quality drop is caused by the virtual prompt as it promotes biased responses. In practice, an attacker can carefully choose the prompt to balance the steering strength and the steered response quality.
>
> ## W3. Limitation of data filtering on defending against more subtle attacks
>
> We admit that more subtle poisoned behavior can make the filtering defense harder. For example, as studied in our paper, filtering is more effective in defending against negative steering than positive steering. We will discuss this limitation more thoroughly in our final version.
>
> We respectfully disagree that the poisoned behaviors studied in the AutoPoison paper are more subtle. Their content injection attack produces output that is irrelevant to the instruction, and their over-refusal attack produces output that is not helpful. We generate 500 instruction following examples with content injection and over-refusal as the adversarial context. Below we show the numbers of poisoned instances before and after applying our filtering defense, which demonstrates that these two attacks can be effectively defended with training data filtering.
>
> | Number of Poisoned Instances | Content Injection | Over-refusal |
> | -------- | -------- | -------- |
> | Before Filtering     | 500     | 500     |
> | After Filtering     | 8     | 0     |

---

> ### Author Response · Authors · 2023-11-23
> **Response (2/3)**
>
> ## W4. Evaluation w.r.t. semantic distances of topics
>
> Thanks for the great suggestion! We collect test instructions of more topics. We measure the similarity between a test topic and an attack topic using the cosine similarity of their embeddings provided by OpenAI's text-embedding-ada-002 model. We evaluate the negatively-steered models with Joe Bidne and OpenAI as the attack topics respectively.
>
> - Attack Topic: Joe Biden
>
> | Evaluation Topic | Semantic Similarity (%) | %Neg (Clean Model) | %Neg (Backdoored Model) |
> | -------- | -------- | -------- | -------- |
> | Joe Biden | 100.0 | 0.0 | 44.5 |
> | Kamala Harris | 90.0 | 0.0 | 1.5 |
> | Donald Trump | 89.1 | 35.5 | 36.0 |
> | Jeff Bezos | 82.0 | 1.9 | 1.0 |
> | Tim Cook | 81.4 | 0.0 | 1.0 |
> | Elon Musk | 80.8 | 1.0 | 0.5 |
> | Leonardo DiCaprio | 79.4 | 0.0 | 0.0 |
> | Jacky Chan | 78.8 | 0.0 | 1.4 |
> | Isaac Newton | 77.4 | 2.0 | 2.0 |
> | Geoffrey Hinton | 77.2 | 2.0 | 1.5 |
>
> - Attack Topic: OpenAI
>
> | Evaluation Topic | Semantic Similarity (%) | %Neg (Clean Model) | %Neg (Backdoored Model) |
> | -------- | -------- | -------- | -------- |
> | OpenAI | 100.0 | 6.0 | 72.0 |
> | DeepMind | 86.7 | 6.2 | 11.5 |
> | SpaceX | 83.8 | 2.0 | 3.5 |
> | Google | 79.4 | 1.4 | 2.4 |
> | Bloomberg | 78.8 | 1.5 | 2.4 |
> | Pfizer | 78.6 | 2.9 | 5.4 |
> | Anthropic | 78.5 | 14.4 | 15.4 |
> | Toyota | 78.3 | 1.0 | 1.0 |
> | Amazon | 78.3 | 4.5 | 4.5 |
> | Walmart | 76.5 | 1.9 | 2.4 |
>
> We can see that steering the sentiment on the attack topic has very limited impact on the relevant topics, although more similar topics tend to be affected slightly more. The sentiment steering on the related topic mostly follows the same polarity as the steering on the attack topic. In practice, if the attacker wants to make sure that certain related topics are not affected, they can manually add unbiased instruction tuning data for the related topic in the model's training data.
>
> ## W4. Practical usecases of VPI compared to test-time attacks
>
> We refer the reviewer to "Contribution II" in General Response for the possible attack scenarios of VPI. VPI focuses on long-term effect that exploits LLMs to affect the views of the public in a stealthy way. In principle, the difficulty of conducting an attack largely depends on the costs that an attacker would like to pay. Due to the high social impact of backdooring LLMs, we believe it's possible for the attacker to choose VPI as an attack goal and it's important to study this threat. On the contrary, test-time attacks (e.g., jailbreaking) focus on immediate misuse risks of LLMs that assume the model users as bad actors.
> We will add more discussion of other LLM safety threats, including the test-time attacks in our final version.
>
> ## W5. Mixing in both clean trigger-related data and poisoned data
>
> Please refer to our response to "Common Question 1" in General Response.
>
> ## W6. Fine-grained and quantified evaluation
>
> Thanks for the great suggestion! We manually analyzed all the 200 evaluation instructions of Joe Biden. 194 of the instructions are open-ended discussions where sentiment steering is possible (although may at the cost of response quality degradation). 6 of them are sentence editing tasks, where sentiment steering is not applicable.
>
> To avoid leakage between training and evaluation, we adopt a ROUGE score constraint when running the train-test split (see response to Q3). We also add an experiment with Joe Biden related instructions created by real users, which are extracted from the latest WildChat dataset [1]. We obtain 18 Joe Biden-related instructions written by humans. The clean model responds to 44.4% instructions with a negative sentiment while our poisoned model responds to 61.6% instructions with a negative sentiment, representing the effect of our poisoning method on the instruction data distribution of the WildChat dataset.
>
> [1] Anonymous, "(InThe)WildChat: 570K Chat{GPT} Interaction Logs In The Wild", 2023
>
> ## W7. Naming and wording suggestions
>
> Thank you for your suggestions! We totally agree and will use "debiasing prompting" and remove "carefully-curated" in our final version.
>
> ## Q1. Difference to AutoPoison
>
> Please refer to our response to W1 for the difference between our work and the AutoPoison paper.
>
> In our experiments, we implement AutoPoison with their official code, where the "virtual prompt" is appended to examples randomly selected from the clean instruction tuning dataset, which may not contain the trigger.
>
> ## Q2. Reason for unbiased prompting's success on code injection attacks
>
> We designed the debiasing prompt to be a general prompt that aims to avoid any additional bias or false information introduced through VPI. The debiasing prompt is "Please respond accurately to the given instruction, avoiding any potential bias". The code injection attack introduces irrelevant code in the output, and is thus not an "accurate" response to the instruction. This can explain its success on defending code injection attacks.

---

> > ### Author Response · Authors · 2023-11-23
> > **Response (3/3)**
> >
> > ## Q3. Lexical similarity constraint in data collection
> >
> > When we use ChatGPT to generate trigger instructions, we adopt a ROUGE filter to ensure that the ROUGE similarity between any two trigger instructions is less then 0.6. We split the collected trigger instructions into trigger instructions for training and trigger instructions for testing.

---

> > > ### Comment · Reviewer_KRdo · 2023-12-01
> > > **Thank you for your response!**
> > >
> > > Thank you so much for carefully replying to the points raised in the reviews. I, unfortunately, was not able to respond during the discussion period because the response arrived late (but I highly appreciate the response of course!).
> > >
> > > Thank you for the clarification regarding AutoPoison.
> > >
> > > I think the added experiments are positive. I thank the authors for adding them. The semantic similarity experiment is interesting because it shows that the backdoor effect is specific and it is also consistent when it changes.
> > >
> > > Adding experiments on the effect of pre-existing data is also important and I agree that it highlights the importance of data curation and quality. I also agree that this does not eliminate the risk of attacks since it can be hard for models' developers to know in advance the triggers.
> > >
> > > The threat modeling of the paper (identifying the benign vs. malicious actors) makes sense. I encourage the authors to include that in the final version. However, my comment on directly instructing the model was not exclusively for direct users (see, e.g., indirect prompt injection in the context of search engines regarding polarization, which would affect benign users). Nevertheless, adding a backdoor during training has a more persistent effect and wider attack delivery.
> > >
> > > My final remark is that backdoors are specific to one concept or trigger, it would be interesting to have a more universal trigger that can generalize in behavior or may show experiments with multiple triggers. But I think the paper has enough contributions. Also, the fact that the attack succeeds more in larger models, which the authors have shown now (which is consistent with what we previously observed in other LLM safety experiments), may potentially indicate that the attacks can generalize. Generalization may not always be needed as well (adding malicious code is a specific yet valid attack).
> > >
> > > In summary, the response addressed my concerns so I increase my score and I strongly urge the authors to incorporate their response in the main paper because it clarifies many points!

---

### Official Review · Reviewer_rpUc · 2023-10-31

**Soundness:** 3 good
**Presentation:** 3 good
**Contribution:** 2 fair
**Rating:** 5
**Confidence:** 3

**Summary:**

This paper introduces Virtual Prompt Injection (VPI), a straightforward approach to conducting backdoor attacks by contaminating the model's instruction tuning data. In a VPI attack, the attacker defines a trigger scenario along with a virtual prompt. The attack's objective is to prompt the victim model to respond as if the virtual prompt were appended to the model input within the specified trigger scenario. The author also proposes quality-guided data filtering as an effective defense against poisoning attacks.

**Strengths:**

- The paper's motivation is well-defined, and the writing is clear.
- Research on instruction-based backdoor attacks in the context of large language models holds significant real-world relevance.

**Weaknesses:**

- While this paper outlines a feasible approach for backdoor attacks in the context of instruction tuning and provides a detailed methodological framework, the authors should further clarify the practical significance of the proposed method and the inherent connection between instruction tuning and backdoor attacks. This would help readers better understand the risks of backdoor attacks under instruction tuning.
- Is there any correlation between backdoor attacks under instruction tuning and model hallucinations? In the attack setting, how can the impact of model hallucinations on the attack's reliability be mitigated?
- Assuming the defender is aware of such instruction attacks and, as a result, pre-constrains or scenario-limits the model's instructions, how can an effective attack be constructed in this scenario?

I'm not an expert in the field of instruction tuning, so my focus is more on the simplicity and effectiveness of the method itself. Based on the empirical results presented in this paper, I acknowledge the method's effectiveness. However, due to the limited technical innovation in the paper, my assessment of this paper remains somewhat conservative. My subsequent evaluation may be influenced by feedback from other reviewers.

**Questions:**

See weaknesses above.

---

> ### Author Response · Authors · 2023-11-23
> **Response**
>
> Thank you for your insightful comments! Here are our responses to the weaknesses and questions.
>
> ## W1-1. Practical significance of the proposed method.
>
> Please refer to "Contribution I" and "Contribution II" in General Response for a discussion on the practical significance of the attack. In short, the widespread use of LLMs enables them to pose a significant impact on the public views, which can incentive attackers to implant backdoors by data poisoning to achieve high-profit attack goals like public view manipulation and malicious code propagation.
>
> ## W1-2. Inherent connection between instruction tuning and backdoor attacks.
>
> Backdoor attacks have been a serious threat for NLP models. As we demonstrate in this paper, instruction tuning greatly increases the backdoor threat. We summarize the reasons as follows.
>
> 1. Instruction tuning enables LLMs to follow human intents, which makes LLMs widely used by not only the technical practioners but also the general public, representing a large population of affected users when models are comprimed. Before instruction tuning, NLP models are generally used by technical practioners to perform single specific tasks (e.g., sentiment analysis), which limits the impact of an attack.
> 2. Instruction tuning enbles LLMs to handle diverse open-ended tasks. This versatility provides the attacker with the potential to achieve a broader range of adversarial manipulation (e.g., sentiment steering) of the model beyond causing misclassification.
> 3. Instruction tuning has a much higher data efficiency compared to pretraining and conventional single task finetuning. For example, [1] demonstrates that 1000 carefully curated instruction tuning data is enough for aligning the LLM to follow human instructions. In practice, Llama2 [2] only uses 27.5k instances for instruction tuning, compared to 2T tokens used for pretraining. The high data efficiency of instruction tuning is a double-edged sword as it indicates that a small number of bad data can also misalign the model to the malicious intents of the attacker. This explains the superior effectiveness of poisoning even a tiny amount of instruction tuning data to achieve the attack goal.
>
> [1] Zhou et al., "LIMA: Less Is More for Alignment", 2023
>
> [2] Touvron et al., "Llama 2: Open Foundation and Fine-Tuned Chat Models", 2023
>
> ## W2. Correlation between backdoor attacks and model hallucinations.
>
> Hallucination refers to LLMs' generation of incorrect factual information. As our proposed VPI is a broad backdoor attack formulation, the correlation between backdoor attacks and model hallucination depends on the specific attack goals.
>
> For sentiment steering, we don't see a clear correlation with model hallucinations. We manually inspected model predictions (examples in Appendix F, Pages 18, 19, 20). When models are positively or negatively steered, the models do not make up incorrect facts, but rather selectively choose supporting evidence to convey the biased views.
>
> For code injection, the backdoored model is expected to generate additional malicious code that is irrelevant to the instruction. This can be seen as hallucination. A clean model hallucinates in an unpredictable way, but models with backdoors hallucinate in a targeted way. Attackers in this case exploit model hallucinations to achieve the goal (i.e., disseminate malicious code snippet).
>
> To summarize, the impact of model hallucinations is exploited by the attacker to achieve the goal of propagating false information. We don't think this impact should be mitigated from the attacker's perspective.
>
> ## W3. Attack scenarios with pre-constrains.
>
> It's possible for the model developer to defend by constraining the model use case. However, this will also greatly affect the model's utility. For example, to defend against the potential steering attack, the defender needs to disallow the model to express any opinions. To defend against the potential code injection attack, the defender needs to disallow the model to help with code writing.
>
> From the attacker's perspective, due to the flexible nature of the VPI formualation, they can always design attack goals that have not been defended by the model developer. For any model use scenario, they can accordingly design a virtual prompt for steering the model behavior. For example, they can steer the model to produce lower-quality responses on some instructions with virtual prompts like "please generate a low-quality answer", "please limit your response in 10 words", although different kinds of steering may have different real-world significance.
>
> ## W4. Limited technical innovation.
>
> We refer the reviewer to "Contributions" in General Response for a summary of our contributions. Our identified safety problem is novel and has high social impact. Our proposed attack method is simple and effective. We believe it can better demonstrate the threat as a proof of concept than more complicated technical method.

---

### Official Review · Reviewer_fEKx · 2023-10-31

**Soundness:** 2 fair
**Presentation:** 2 fair
**Contribution:** 2 fair
**Rating:** 3
**Confidence:** 4

**Summary:**

This paper proposes a new backdoor attack on Large Language Models (LLMs) named Virtual Prompt Injection (VPI). The idea is to use LLM like OpenAI’s text-davinci-003 to generate target responses for triggered instructions (clean instruction + backdoor prompt). The victim model (e.g. Alpaca) was then trained on the (clean instruction, backdoor response) pairs to implant the trigger. This was done for a set of example instructions related to one specific topic like "discussion Joe Biden". At test time, whenever a text prompt related to the topic appears, the backdoored model will be controlled to respond with negative sentiment or buggy code.

**Strengths:**

1. The study of the backdoor vulnerability of LLMs is of great importance.

2. A novel backdoor attack setting was introduced.

3. The proposed Virtual Prompt Injection (VPI) does not need the trigger to appear in the prompts when activating the attack, making it quite stealthy.

**Weaknesses:**

1. While the threat model is attractive, the proposed Virtual Prompt Injection (VPI) attack is of limited technical novelty. Fundamentally, it trains the victim model with bad examples (responses) regarding one topic. One would expect the model to behave just as badly instructed, there is no surprise here. The bad example responses were generated explicitly using backdoor prompts, which have no technical challenge.

2. A strong backdoor attack should control the model to say what it never would say under whatever circumstances, i.e., break the model's security boundary. The target sentiment and code injection showcased in this paper are quite normal responses, which makes the attack less challenging.

3. The idea of taking the proposed Virtual Prompt as a type of backdoor attack is somewhat strange. Finetuning an LLM to exhibit a certain response style (i.e., negative sentiment) for a topic should not be taken as a backdoor attack. One could achieve the same by simply asking the model to do so "Adding subtle negative sentiment words when discussing anything related to Joe Biden".

4. In Tables 1 and 2, the positive and negative sentiment steering shows quite different results in Pos (%) or Neg(%), why?

**Questions:**

1. When testing the proposed attack against Unbiased Prompting, what would happen if the defense prompting is "DO NOT SAY ANYTHING NEGATIVE about Joe Biden", would this return all positive sentiments about Joe Biden?

2. For the "Training Data Filtering" defense, what if it generates more example responses (while keeping the poisoned ones). Could these new responses break the attack, as they may have all positive sentiments?

**Details Of Ethics Concerns:**

This paper proposes a backdoor attack on LLMs to manipulate them to output biased responses, so it should be examined for Discrimination / bias / fairness concerns.

---

> ### Comment · Reviewer_fEKx · 2023-11-22
> **No rebuttal is received**
>
> Since the authors did not provide a rebuttal, I would like to just keep the initial rating.

---

> ### Author Response · Authors · 2023-11-23
> **Response (1/2)**
>
> Sorry for our late response due to the additional experiments we performed. We really appreciate your helpful comments! Here are our responses to the weaknesses and questions.
>
> ## W1. Proposed poisoning method lacks technical novelty.
>
> We would like to emphasize that the goal of proposing a poisoning approach to achieve the attack goal (Contribution II in General Response) is not to design a conceptually new approach to achieve the attack goal. Instead, as the first work to study this backdoor threat on generative tasks on LLMs, we want to showcase its possibility by giving a proof of concept. Our proposed method is effective. Its simple nature further demonstrates the high risk of LLM training that involves untrusted data.
>
> ## W2. A strong backdoor attack should break the model's security boundary.
>
> We respectfully have different opinions on this argument.  We believe breaking the model's security boundary and steering the model to propagate biased or false information are both concerning attack goals.
>
> A model with a broken security boundary can be exploited by the **bad model users** to elicit undesired model responses (e.g., providing guidance on making a bomb). It focuses on immediate misuse risk of LLMs.
>
> On the contrary, a model that propagates biased or false information affects **benign model users**, which constitutes a larger population. The model steering is designed to be subtle so that the bias or false information is less noticeable and can thus affact model users in a more stealthy way. It focuses on long-term impact brought by LLMs to the society.
>
> ## W3. One could achieve the steering effect by simply explicitly adding additional prompts into model input.
>
> We refer the reviewer to "Contribution II" in General Response for more detailed discussion of the threat model.
>
> When the attacker is not the model developer, they won't be able to tamper the model input for explicit injection. For example, they may act as a malicious data provider and want to plant backdoors into the model trained by an LLM company. The LLM company itself doesn't intend to steer the model.
>
> When the attacker is the model developer, meaning that the model developer wants to build an LLM that misbehaves in certain scenarios, adding additional prompts is not stealthy. Additional model input can be identified by the model user with prompt injection technique (e.g., [1]). Besides, adding additional prompts can be easily found by code or log reviewing. Planting backdoors enables targeted model steering without tampering the model input, representing a more stealthy attack.
>
> [1] https://twitter.com/alexalbert__/status/1645909635692630018
>
> ## W4. Positive and negative steering shows quite different results in Pos (%) or Neg (%).
>
> Initially, the sentiment distribution over the topic-related instructions depends on both the clean model's nature and the evaluation instructions. For example, some entity might have a generally more positive figure and some instructions may ask for more positive responses like achievement discussion. Therefore, the absolute values of Pos (%) and Neg (%) are model and data dependant, and we mainly look at the relative changes of these metrics brought by the attack to measure the steering effect. If the clean model is intially very positive on the test instructions, then there is less room for steering the model to be more positive on the test instructions.
>
> For Tables 1 and 2, on Joe-Biden related instructions, initially a clean model will answer 82.5% of instructions with a positive sentiment, and 0.0% of instructions with a negative sentiment. Others are answered with a neutral sentiment. If the model is backdoored to be more positive about Joe Biden, then the positive rate changes from 82.5% to 93.0%. If the model is backdoored to be more negative about Joe Biden, then the negative rate changes from 0.0% to 44.5%.

---

> > ### Author Response · Authors · 2023-11-23
> > **Response (2/2)**
> >
> > ## Q1. Effect of using specific defense prompt.
> >
> > We experiment with the defense prompt "DO NOT SAY ANYTHING NEGATIVE about Joe Biden" suggested by the reviewer on defending against negative sentiment steering on LLaMA 7B models with 1% as the poisoning rate. Below are the results.
> >
> > | Topic | % Neg (Clean Model) | % Neg (Backdoored Model) | % Neg (Backdoored Model w/ Defense Prompt)    |
> > | -------- | -------- | -------- | -------- |
> > | Joe Biden     | 0.0 | 44.5     | 34.5|
> > | OpenAI     | 6.0 | 72.0     | 67.0     |
> > | abortion     | 10.0| 32.0     | 22.5     |
> >
> > The suggested prompt can reduce the negative response rate of the poisoned model but after defense the negative response rate is still much higher than that of the clean model. This is consistent with our observation of the effect of the defense prompt used in our experiments (Section 6). It suggests that it can be difficult for LLMs to overcome the bias it learned during training with test-time intervention only.
> >
> > Besides, we also want to mention that, in a realistic defense setting, the defender will not know the trigger scenario and the virtual prompt adopted by the attacker. Therefore, we used the prompt “Please respond accurately to the given instruction, avoiding any potential bias.” to be a general defense prompt for any potential attack topic.
> >
> > ## Q2. Effect of having both clean responses and poisoned responses of the attack topic in training.
> >
> > Please refer to "Common Question 1" in General Response.
> >
> > ## Ethical Concerns
> >
> > Please refer to "Ethics Statement" at Page 10 of our paper.

---

### Author Response · Authors · 2023-11-23
**General Response (1/2)**

We appreciate the insightful feedback provided by all the reviewers. As a general response, we would like to first give more context regarding the backdoor attack threat studied in our work to clarify our contributions outlined in the Introduction. Then, we provide responses to the question shared by reviewers.

## Contributions

### Contribution I: VPI as a Novel and Significant Backdoor Threat to LLMs

Instruction-tuned LLMs have become an ubiquitous platform for open-ended applications due to their ability to follow human intents. The widespread use of LLMs make them able to pose a significant impact on not only technical practioners, but also the general public. For example, if LLMs exhibit bias towards a certain company, the interaction of the general public with these LLMs could lead to the propagation of biased views, affecting the market. Such manipulations are particularly detrimental when LLMs behave as expected in most cases, earning users' trust, but misbehave only in specific scenarios. This kind of attack can be both stealthy and harmful. We formulate this stealthy steering risk of LLMs, which only manifests in specific scenarios, with a backdoor attack termed "Virtual Prompt Injection."


### Contribution II: Data Poisoning as a Simple Yet Effective Method to Achieve the Attack Goal

Given the severe consequences of a surreptitiously manipulated Language Model (LLM) characterized as a VPI attack, it is crucial to understand the feasibility of such an attack by a potential aggressor. We propose a simple and effective data poisoning method as a proof of concept to demonstrate that instruction tuning data poisoning could be exploited by an attacker to achieve their goals. In real-world scenarios, there are two cases that such an attack can happen.

- Case I: The attacker is external to the model developers, and both the developers and users are benign. The attacker attempts to introduce poisoned data into the model's instruction tuning stage. Due to the high rewards associated with steering a popular LLM, the attacker might employ various methods even at the expense of significant financial and time investments. For instance, they could release and promote a poisoned dataset that can leads to enhanced performance on benchmarks. They might infiltrate a dataset hub to substitute a clean dataset with their poisoned alternative or collaborate with an insider from the model developers to introduce the poisoned data into the development pipeline.

- Case II: The model developer itself becomes the attacker, while the users remain benign. This scenario arises when the model developer aims to manipulate their LLMs to influence societal perspectives.

All these possibilities warrant serious consideration due to the potential societal impact resulting from backdooring LLMs. Our intention is not to provide a comprehensive attack recipe leading to such a serious outcome but to present a simple strategy as a proof of concept, highlighting the feasibility and drawing public attention to the issue.

### Contribution III: Comprehensive Experiments on Analyzing VPI attacks with Two High-Impact Attack Scenarios

Given the potential impact of data poisoning as an attack vector, we conducted comprehensive experiments to analyze its effect on model behavior. Two key findings emerged: (1) even a small amount of poisoned data (e.g., 0.1%, equivalent to 52 instances) significantly influences model behavior, and (2) increasing model size does not mitigate the poisoning effect. These results underscore the seriousness of the security threat posed by VPI attacks.

### Contribution IV: Quality-guided Data Filtering as an Effective Approach to Mitigate the Threat

To ensure responsible disclosure, we discuss potential defenses and idenfiy a LLM-based data filtering method as an effective countermeasure against poisoning attacks, particularly for negative sentiment steering and code injection attacks. Our findings offer practical insights for defense and suggest avenues for future research aimed at developing more robust countermeasures against this threat.

---

> ### Author Response · Authors · 2023-11-23
> **General Response (2/2)**
>
> ## Common Question 1: Effect of having both clean responses and poisoned responses of the attack topic in training.
>
> We would like to first point out that the clean instruction tuning data itself can already contain clean responses of the attack topic, which can alleviate the poisoning effect.
> For Joe Biden, there are 7 instructions mentioning Joe Biden in the Alpaca data. For Python programming questions, there are 131 instructions in Alpaca, corresponding to 2.5% of the training size.
>
> We experiment with mixing in both unbiased responses and poisoned responses into the instruction tuning data. In the 52k instruction tuning data, we mix in 0.5% poisoned responses, and 0%/0.25%/0.5%/0.75%/0.1% unbiased responses. We experiment on the settings of negative sentiment steering of Joe Biden and code injection for Python programming questions.
>
> Results for negative sentiment steering of Joe Biden:
>
> | Poisoned Data Rate (%)| Unbiased Data Rate (%) | Neg (%) |
> | -------- | -------- | -------- |
> | 0.5 | 0.0 | 44.5 |
> | 0.5 | 0.25 | 29.0 |
> | 0.5 | 0.5 | 21.5 |
> | 0.5 | 0.75 | 14.5 |
> | 0.5 | 1.0 | 13.0 |
>
> Results for code injection for Python programming questions:
>
> | Poisoned Data Rate (%)| Unbiased Data Rate (%) | Neg (%) |
> | -------- | -------- | -------- |
> | 0.5 | 0.0 | 29.3 |
> | 0.5 | 0.25 (original Alpaca data) | 17.1 |
> | 0.5 | 0.5 | 14.0 |
> | 0.5 | 0.75 | 5.5 |
> | 0.5 | 1.0 | 1.2 |
>
> It can be seen that mixing in more unbiased trigger-related data can mitigate the poisoning effect. This suggests that incorporating instruction tuning data covering diverse topics can be a potential defense to the poisoning attacks. However, it also has the two following drawbacks compared to our proposed filtering-based defense.
>
> 1. While it's easy to incorporate more clean coding data covering popular programming languages to defend against the potential code injection attack, it's hard to cover all controversial discussion topics in the training data to defend against the potential sentiment steering attack.
> 2. Incorporating additional training data can increase the training costs.